# Effects of Exposure of PHMG-p, a Humidifier Disinfectant Component, on Eye Dryness: A Study on a Rat Model Based on ^1^H-NMR Metabolomics

**DOI:** 10.3390/ijms26178660

**Published:** 2025-09-05

**Authors:** Jung Dae Lee, Hyang Yeon Kim, Soo Bean Oh, Hyeyoon Goo, Kyong Jin Cho, Gi-Wook Hwang, Suhkmann Kim, Kyu-Bong Kim

**Affiliations:** 1College of Pharmacy, Dankook University, Cheonan 31116, Republic of Korea; ljd0734@nate.com (J.D.L.); festivalkim@gmail.com (H.Y.K.); 2Center for Human Risk Assessment, Dankook University, Cheonan 31116, Republic of Korea; 3College of Medicine, Dankook University, Cheonan 31116, Republic of Korea; ohsu9212@naver.com (S.B.O.); ghy1204@hanmail.net (H.G.); perfectcure@dankook.ac.kr (K.J.C.); 4Division of Environmental and Health Sciences, Faculty of Pharmaceutical Sciences, Tohoku Medical and Pharmaceutical University, Sendai 981-8558, Japan; hwang@tohoku-mpu.ac.jp; 5Department of Chemistry, Center for Proteome Biophysics, and Chemistry Institute for Functional Materials, Pusan National University, Busan 46241, Republic of Korea; suhkmann@pusan.ac.kr

**Keywords:** PHMG-p, eye dryness, NMR, metabolomics, biomarker

## Abstract

Polyhexamethylene guanidine phosphate (PHMG-p), a widely used disinfectant component in household humidifiers, has been implicated in various health issues, including pulmonary toxicity. Many people use humidifiers to improve dry eye disease (DED). The current study was performed to elucidate the effect of PHMG-p on eye dryness in a rat model using metabolomics. Male Sprague Dawley rats were exposed to PHMG-p (0.1% and 0.3%) following a previously established DED induction model using scopolamine hydrobromide and desiccation stress. Ocular surface damage was assessed using corneal fluorescein staining, tear volume measurement, and tear break-up time (TBUT). Plasma and urine samples were analyzed using ^1^H-NMR-based metabolomics to identify metabolic alterations associated with PHMG-P-p exposure and DED pathogenesis. PHMG-p exposure exacerbated DED symptoms, as evidenced by a significant reduction in tear volume, shorter TBUT, and increased corneal damage compared to the control group. Metabolomic profiling identified distinct metabolic changes in PHMG-p-exposed groups, including alterations in glutamate, glycine, citrate, and succinate metabolism. These metabolic changes correlated with increased levels of inflammatory cytokines such as IL-1β, IL-6, and TNF-α in the corneal and lacrimal gland tissues. Our findings suggest that PHMG-p exposure contributes to DED pathophysiology by inducing metabolic disturbances and inflammatory responses in the ocular surface. This study highlights the need for further investigation into the potential risks of PHMG-p exposure on ocular health and provides novel insights into the metabolic underpinnings of DED.

## 1. Introduction

Dry eye disease (DED), also known as dry eye syndrome or keratoconjunctivitis sicca, is a multifactorial and chronic ocular condition characterized by a loss of tear film homeostasis [1]. It affects the ocular surface and is associated with symptoms such as burning, itching, heaviness, eye fatigue, dryness, facial flushing, and blurred vision. DED occurs when the eyes fail to maintain an adequate tear layer, essential for lubricating and nourishing the ocular surface. This leads to increased osmotic pressure in the tear film, resulting in irritation and inflammation, further exacerbated by the release of pro-inflammatory mediators [2]. With an estimated prevalence of 5–50% globally, DED is one of the most common ophthalmic disorders [3]. DED has a variety of causes and risk factors. Age is a significant contributor, with older individuals being more susceptible. Certain medications, such as antihistamines, antihypertensives, and antidepressants, have been associated with DED development. Autoimmune diseases like Sjögren’s syndrome, environmental factors such as low humidity and high airflow, and meibomian gland dysfunction, which affects the lipid layer of the tear film, are also prominent causes [1]. Symptoms include stinging or burning sensations, pressure, a gritty feeling, redness, blurry vision, excessive tearing, and difficulty wearing contact lenses, all of which significantly affect the quality of life [1,4].

DED is typically diagnosed using methods such as ocular surface staining, non-invasive evaluation of tear break-up time (TBUT), and measurement of osmotic pressure [5]. The pathophysiology of DED involves inflammation of the ocular surface and increased osmotic pressure of the tear film, which play critical roles in disease progression [6,7]. Common features include inflammation of the ocular surface and major tear glands. Key inflammatory markers such as human leukocyte antigen-DR (HLA-DR) and interleukin (IL)-1β have been detected in the conjunctiva, contributing to the apoptosis or cytotoxicity of conjunctival epithelial cells [8,9,10,11]. Increased tear osmotic pressure has also been linked to the overexpression of pro-inflammatory cytokines, including tumor necrosis factor-alpha (TNF-α), IL-1, and IL-6, perpetuating the inflammatory cascade in DED [12,13,14].

TNF-α, IL-1β, and IL-6 are central to DED pathogenesis. These cytokines are significantly elevated in the tear film and ocular tissues of DED patients, correlating with disease severity and maintaining the chronic inflammatory state [15,16]. TNF-α and IL-6 interact synergistically, amplifying the inflammatory response and stimulating antigen-presenting cell maturation and T-cell function, which further sustain the inflammatory cycle. Elevated IL-1β levels inhibit neurotransmitter secretion, reducing tear production and exacerbating ocular surface damage [16]. Furthermore, TNF-α is pivotal in orchestrating inflammation, with topical TNF-α blockers shown to suppress inflammation in the cornea and lacrimal glands by reducing IFN-γ, IL-21, and IL-6 expression [15,17].

The therapeutic importance of targeting these cytokines is evident in the effectiveness of anti-inflammatory treatments. For instance, cyclosporine A combined with artificial tears significantly reduces TNF-α, IL-1β, and IL-6 levels, improving clinical outcomes in DED patients [15,16]. These findings highlight the critical role of pro-inflammatory cytokines in the pathogenesis and progression of DED and underscore the potential of targeted therapies in managing this disease.

According to recent data, the number of patients visiting hospitals for DED in the Republic of Korea ranges between 2.3 million and 2.5 million annually. As of 2022, women accounted for 66.5% of the total patients, approximately twice the number of men. By age group, patients in their 60s represented the largest proportion at 19.4%, followed by those in their 50s (19.1%) and 40s (15.1%). These statistics highlight that DED is a highly prevalent condition in the Republic of Korea, particularly among women and middle-aged to elderly individuals [18].

Polyhexamethylene guanidine phosphate (PHMG-p) is a guanidine derivative widely used as a biocidal disinfectant due to its fungicidal and bactericidal properties against Gram-positive and Gram-negative bacteria. In the Republic of Korea, PHMG-p was commonly utilized as a disinfectant in household humidifiers to prevent microbial contamination [19,20]. However, in the Republic of Korea, the use of PHMG-p in household humidifiers led to a public health crisis, prompting investigations into its toxicity and long-term effects [21]. PHMG-p exposure was found to primarily target the respiratory system, causing severe health risks, which has driven extensive research across various fields. Studies on the toxicity of PHMG-p have been conducted in areas such as respiratory health, cytotoxicity, immunology, hematology, histology, and omics [22,23,24,25,26,27,28]. Due to its harmful effects, ongoing research continues to elucidate the mechanisms of PHMG-p-induced damage and explore potential therapeutic strategies for affected individuals.

The hypothesis that the increase in DED prevalence might be associated with the use of humidifier disinfectants was supported by several observations: (1) the use of humidifiers was often recommended for the treatment of DED [29], (2) while the prevalence of DED has been steadily increasing before and after the discontinuation of humidifier disinfectants in 2011, there was an unexpected decline in 2013, and (3) women, who were more likely to engage in indoor activities, exhibited a prevalence rate more than twice that of men, who were more exposed to outdoor activities [30]. This is contrary to the expected trend if factors such as yellow dust or fine particulate matter were the primary causes of DED. Therefore, investigating the correlation between humidifier disinfectants and DED is both meaningful and important.

Metabolomics has emerged as a powerful tool in disease diagnosis and treatment, offering valuable insights into pathophysiological mechanisms and potential therapeutic targets [31,32]. By comprehensively characterizing metabolites within biological systems, metabolomics provides a unique perspective on disease processes and treatment outcomes, enabling a deeper understanding of the molecular basis of various conditions [33,34,35]. In the field of disease diagnosis, metabolomics has demonstrated significant potential across a broad spectrum of applications. One of its most promising contributions is the identification of biomarkers for early detection of diseases such as cancer, diabetes, and cardiovascular disorders, allowing for timely intervention and improved patient outcomes [34,36,37]. Clinically, metabolomics has already been implemented in newborn screening programs, enabling the detection of over 50 inherited metabolic disorders, thereby facilitating early treatment and prevention of severe complications [38,39]. Moreover, its potential in infectious disease diagnosis is gaining recognition, as it may allow for the direct detection of pathogens or the identification of host-response biomarkers specific to particular infections [33,40,41]. In the context of complex diseases, metabolomics has provided novel insights into the metabolic underpinnings of conditions such as diabetes, Alzheimer’s disease, atherosclerosis, and cancer, uncovering biomarkers and metabolic pathways previously unrecognized [42,43,44,45].

Moreover, many studies have employed metabolomics to identify biomarkers and understand the disease mechanisms of DED. Tear metabolomics has been a focal point of research due to the direct relevance of tear composition to ocular surface health. A cross-sectional study involving 113 participants (85 with DED and 28 controls) analyzed reflex tears using UPLC-Q/TOF-MS/MS, revealing 48 metabolites associated with DED incidence [42]. These metabolites showed variations across age groups and were primarily involved in glucose metabolism, amino acid metabolism, and glutathione metabolism. Furthermore, metabolic changes were correlated with clinical indicators such as the Ocular Surface Disease Index (OSDI) and fluorescein breakup time [42]. Serum metabolomics has also provided valuable insights, as evidenced by a large-scale study involving 2819 subjects from the Twins UK cohort [43]. This study identified 222 serum metabolites and highlighted the critical role of androgen metabolism in DED development, particularly among females [43]. Additionally, combined tear and saliva metabolomics studies have advanced the understanding of evaporative DED, particularly in female patients. Recent research identified 56 metabolites in tears that significantly differed between DED patients and controls, with these metabolites linked to meibum composition, antioxidative properties, and the ocular microbiome [44]. Saliva analysis revealed lower levels of hypotaurine in patients with tear film instability, suggesting a potential link between systemic metabolism and ocular health [44]. Animal model studies have further contributed to the field, with scopolamine-induced DED models used to analyze plasma and urine metabolites [45,46]. These studies aimed to identify potential biomarkers and elucidate metabolic changes associated with DED [45,46]. In therapeutic research, in situ metabolomics using MALDI-MSI has identified potential treatment targets [47]. A study published in Nature (2025) demonstrated increased glutamine levels in the cornea following combined mesenchymal stem cell and thymosin beta-4 therapy. This research identified glutaminase 1 (GLS1) as a potential therapeutic target for DED, paving the way for novel treatment approaches [47].

Therefore, in previous studies, we identified biomarkers of dry eye disease (DED) using ^1^H-NMR in a rat model [45]. Building on this foundation, we aimed to investigate the correlation between PHMG-p, a component of humidifier disinfectants, and its role in inducing or exacerbating DED.

## 2. Results

### 2.1. Corneal Fluorescein Staining and Corneal Damage

On day 13, corneal epithelial damage was assessed using corneal fluorescein staining. Both eyes of each rat in the groups (*n* = 6) were examined, and representative corneal fluorescein staining images from each group on day 13 after DED induction are shown in Figure 1A. The corneal damage scores were 0.68 ± 0.49 and 1.75 ± 0.75 in the control group and the DED group, respectively. The scores for the PC group, DED + PHMG-p 0.1% group, and DED + PHMG-p 0.3% group were 2.25 ± 0.75, 2.08 ± 0.29, and 2.08 ± 0.34, respectively. Additionally, the PHMG-p 0.1% group and the PHMG-p 0.3% group showed scores of 2.25 ± 0.75 and 1.42 ± 0.34, respectively. The corneal damage scores were significantly higher in all groups compared to the control group (Figure 1B).

### 2.2. Tear Volume

The mean tear volumes across the experimental groups are summarized as follows. In the control group, the tear volume was 0.24 ± 0.08 μL. In the DED group, the tear volume was reduced to 0.13 ± 0.05 μL, confirming a significant decrease in tear production and the successful induction of DED compared to the control group. In the PC group (DED + BAC), treated with benzalkonium chloride (0.1%) after DED induction, the tear volume was further reduced to 0.11 ± 0.04 μL, showing an exacerbation of DED symptoms. The groups treated with 0.1% or 0.3% PHMG-p after DED induction exhibited mean tear volumes of 0.12 ± 0.07 μL and 0.15 ± 0.02 μL, respectively, while the groups treated with PHMG-p (0.1% or 0.3%) alone had mean tear volumes of 0.07 ± 0.03 μL and 0.30 ± 0.10 μL, respectively. All groups, except for the group treated with 0.3% PHMG-p alone, showed significant differences compared to the control group (Figure 1C).

### 2.3. Tear Break-Up Time (TBUT)

The TBUT results of the control group and the DED group were 7.63 ± 2.64 s and 5.08 ± 1.92 s, respectively, after 13 days of exposure to desiccation stress. The PC group showed a TBUT of 3.79 ± 0.70 s. The groups treated with 0.1% or 0.3% PHMG-p after DED induction had TBUT values of 3.81 ± 1.37 s and 5.40 ± 1.12 s, respectively, while the groups treated with PHMG-p alone had TBUT values of 5.57 ± 0.69 s and 6.32 ± 0.50 s, respectively. Compared to the control group, TBUT was significantly reduced in all groups except for the group treated with only PHMG-p 0.3% (Figure 1D). TBUT was calculated using the mean of three measurements.

### 2.4. Conjunctival Goblet Cell Counts

The total number of goblet cells in the control group was 89 ± 8.66 cells/mm^2^, while in the DED group, the goblet cell count was significantly reduced to 17 ± 13.75 cells/mm^2^. The PC group exhibited a further reduction in goblet cell count, with 3 ± 5.20 cells/mm^2^. In the DED + PHMG-p 0.1% and DED + PHMG-p 0.3% groups, the goblet cell counts were 0.33 ± 0.58 cells/mm^2^ and 38 ± 4.37 cells/mm^2^, respectively. The PHMG-p 0.1% group showed a goblet cell count of 3 ± 3.61 cells/mm^2^, while the PHMG-p 0.3% group demonstrated 75.5 ± 7.00 cells/mm^2^. Compared to the control group, the goblet cell count was significantly reduced in all groups except for the group treated with only PHMG-p 0.3%. Representative histological findings of goblet cell counts in each group are presented in Figure 2.

### 2.5. Inflammatory Cytokine Concentrations in the Cornea

The expression levels of IL-6, IL-1β, and TNF-α in the corneal tissue were assessed and are shown in Figure 3A. The DED group exhibited a significant increase in all three pro-inflammatory cytokines compared to the control group. The PC group also showed elevated cytokine levels, although the increase was less pronounced compared to the DED group for some markers. In the groups treated with PHMG-p after DED induction, IL-6 and TNF-α levels were significantly elevated in the DED + PHMG-p 0.3% group compared to the control group. IL-1β levels were similarly increased in the DED + PHMG-p 0.3% group, while the DED + PHMG-p 0.1% group exhibited moderately elevated cytokine levels. In the PHMG-p-only groups, IL-6, IL-1β, and TNF-α were slightly increased in the PHMG-p 0.1% group but remained lower than those observed in the DED + PHMG-p groups. The PHMG-p 0.3% only group showed relatively lower cytokine expression levels compared to DED + PHMG-p 0.3%. Representative histological images of IL-1β, IL-6, and TNF-α staining are presented in Figure 3B. Immunohistochemical staining revealed positive immunoreactivity for IL-6, IL-1β, and TNF-α in the DED and DED + PHMG-p groups (indicated by red arrows). These groups exhibited dense chromogenic deposition, suggesting enhanced inflammatory signaling in the corneal tissue. In contrast, weaker or minimal staining was observed in the control and PHMG-p-only groups, indicating relatively low basal cytokine expression. The staining intensity and distribution correspond well with the quantified data shown in Figure 3A, highlighting the pro-inflammatory effects of DED and PHMG-p exposure at the tissue level.

### 2.6. Plasma NMR Profile

NMR spectra for plasma samples of control or treated groups were obtained. The spectral region of δ0.0–10 was segmented into regions of 0.04 ppm width, providing 250 integrated regions in each NMR spectrum for plasma samples. Visual examination of NMR spectra displayed different intensities of several metabolites between groups. The spectral binning data were obtained through NMR analysis of S–D rat plasma samples.

The global profiling data exhibited clear clustering among the control, DED, PC, and PHMG-p-treated groups on the principal component analysis (PCA). Additionally, the orthogonal projections to latent structures discriminant analysis (OPLS-DA) data showed a clear separation of clusters between the groups, as analyzed using the SIMCA-P multivariate analysis program (Figure 4A,B). A total of 41 endogenous metabolites were identified using Chenomx NMR Suite ver. 8.3 (Chenomx Inc., Edmonton, AB, Canada) in plasma samples of control, DED, PC group and PHMG-p-treated groups. The score plots of PCA and OPLS-DA in plasma target profiling demonstrated clearly discriminable clustering between groups. The VIP showed sorting of endogenous metabolites in the order of contribution to the separation of clustering (Figure 5). Significant metabolites were selected according to a VIP value of more than 0.5, which determined meaningfully important metabolites (Figure 5C). In addition, we compared the results with the DED-related metabolites identified in our previous study [44]. In a previous study, the major metabolites associated with DED were identified as 1,3-dimethylurate, 2-hydroxyisobutyrate, alanine, citrate, creatine, glucose, glutamate, lactate, N-nitrosodimethylamine, and succinate. A total of 20 major plasma metabolites were selected in the present study, including 2-hydroxyisobutyrate, 3-hydroxybutyrate, acetate, alanine, arginine, citrate, creatine, glucose, glutamate, glutamine, glycerol, glycine, isoleucine, lactate, leucine, pyruvate, serine, succinate, threonine, and valine. Among them, eight metabolites—2-hydroxyisobutyrate, alanine, citrate, creatine, glucose, glutamate, lactate, and succinate—were identified, which were the same as those reported in the previous study [45]. Figure 6 shows the changes in metabolites for each group.

After performing pathway analysis for the selected metabolites using the MetaboAnalyst 6.0 program (http://www.metaboanalyst.ca (accessed on 11 November 2024)) [48], 21 metabolic pathways were predicted in plasma samples. Among these pathways, glycine and serine metabolism, glutamate metabolism, arginine and proline metabolism, glutathione metabolism, the citric acid cycle, amino sugar metabolism, and fatty acid biosynthesis were identified (Table 1). Figure 7 illustrates the expected changes in plasma metabolic pathways in the PHMG-p-treated group after DED induction.

### 2.7. Urinary NMR Profile

NMR spectra for urine samples of control or treated groups were obtained. The spectral region of δ0.0–10 was segmented into regions of 0.04 ppm width, providing 250 integrated regions in each NMR spectrum for urine samples. Visual examination of NMR spectra displayed different intensities of several metabolites between groups.

The global profiling data showed clear separation of clustering among the control, DED, PC, and PHMG-p-treated groups on the par-scale of PCA and OPLS-DA models (Figure 4C,D). A total of 88 endogenous metabolites were identified using Chenomx NMR Suite ver. 8.3 (Chenomx Inc., Edmonton, AL, Canada) in the urine samples of each group. PCA and OPLS-DA score plots in urinary target profiling exhibited clear discrimination of clustering between groups (Figure 8). VIP demonstrated sorting of endogenous metabolites in the order of contribution to separation of clustering. Significant metabolites were selected according to a VIP value of more than 0.5, which determined meaningfully important metabolites (Figure 8C).

In a previous study, 26 urinary metabolites were screened in the DED group, among which phenylalanine, phenylacetate, pantothenate, glycine, succinate, methanol, valine, propylene glycol, histidine, threonine, lactate, and acetate showed significant differences [45]. In this study, a total of 36 major urinary metabolites were identified, including 1-methylnicotinamide, 2-oxoglutarate, 3-hydroxybutyrate, 4-hydroxyphenylacetate, 4-pyridoxate, acetate, acetone, allantoin, asparagine, cis-aconitate, citrate, dimethylamine, ethanol, ethylene glycol, formate, glucose, glutamine, glycine, hippurate, lactate, lysine, N6-acetyllysine, N-acetylglycine, N-methylhydantoin, N-phenylacetylglycine, ornithine, phenylalanine, propylene glycol, sarcosine, succinate, succinylacetone, taurine, trigonelline, trimethylamine, trimethylamine N-oxide, and uracil. Among them, 12 metabolites were identical to those reported in previous studies: acetate, cis-aconitate, glycine, lactate, N6-acetyllysine, N-methylhydantoin, phenylalanine, propylene glycol, sarcosine, succinate, taurine, and trimethylamine. Figure 9 shows the changes in metabolites for each group.

Pathway analysis for the selected metabolites in urine samples was also performed in the same manner as for plasma samples, and 26 metabolic pathways were predicted. The predicted metabolic pathways include the citric acid cycle, glutamate metabolism, aspartate metabolism, arginine and proline metabolism, glycine and serine metabolism, amino sugar metabolism, fatty acid biosynthesis, and taurine and hypotaurine metabolism (Table 2). The expected changes in urinary metabolic pathways in the PHMG-p-treated group after DED induction are illustrated in Figure 10.

## 3. Discussion

In our previous study, we induced DED in rats and used a metabolomics approach to develop biomarkers and elucidate the mechanisms underlying DED [45,46]. The present study investigated the effects of polyhexamethylene guanidine phosphate (PHMG-p), a humidifier disinfectant component, on DED using a rat model and metabolomic analysis.

Several studies have investigated the ocular effects of polyhexamethylene guanidine (PHMG), highlighting its potential toxicity and concentration-dependent impact on eye health. A study by Lee et al. (2021) examined the role of fibrosis as a manifestation of PHMG toxicity in the eye using Statens Seruminstitut Rabbit Cornea (SIRC) cells [49]. The study demonstrated that PHMG exposure led to an increased expression of fibrosis-related biomarkers, including TGF-β, α-SMA, MMPs, and TIMPs, at both the gene and protein levels. Furthermore, oxidative stress levels were significantly elevated in PHMG-treated cells, supporting the hypothesis that PHMG can induce fibrosis in the cornea. Another study by Park et al. (2019) evaluated the eye irritation potential of PHMG using a reconstructed human cornea-like epithelium model (EpiOcular™) [50]. The results indicated that raw PHMG materials, with an active ingredient concentration of 26%, were classified under UN GHS Category 1 (serious eye damage) or Category 2 (eye irritation). However, when PHMG solutions were diluted to 0.13% or lower, no significant eye irritation was observed at the tested concentrations. A more recent systematic review by Ivanov et al. (2024) provided a comprehensive assessment of PHMG’s toxicity and safety through various exposure routes, including ocular contact [51]. The review concluded that PHMG-p solutions below 0.13% appear to be safe for the human corneal epithelium. However, despite this, even low concentrations of PHMG have been associated with corneal fibrosis, as demonstrated in animal studies. These findings suggest that while diluted PHMG solutions may not cause immediate eye irritation, chronic exposure could still pose a long-term risk to corneal health, particularly through fibrotic changes and oxidative stress-related damage. Although several studies have reported on PHMG and ocular health, no studies have directly investigated its link to DED. Meanwhile, since humidifiers were often recommended for the treatment of DED, this study aimed to explore the potential role of PHMG-p, a component of humidifier disinfectants, in aggravating ocular surface damage and metabolic disorders.

In the present study, changes in conjunctival goblet cell (CGC) density and corneal inflammatory cytokines (IL-6, IL-1β, and TNF-α) were observed, aligning with previous findings. Several studies have investigated the relationship between DED and alterations in CGC numbers, emphasizing their critical role in maintaining tear film stability through mucin secretion. The loss of goblet cells has been strongly implicated in the pathogenesis of DED. Experimental mouse models of DED have demonstrated that DED induction promotes the migration of CD4^+^ T cells and IFN-γ^+^ cells into conjunctival goblet cell zones, leading to progressive goblet cell loss [52]. This immune-mediated response contributes to tear film instability and ocular surface damage commonly observed in DED. More recently, a 2023 study utilized moxifloxacin-based fluorescence microscopy (MBFM) to noninvasively assess goblet cell changes in DED-induced mice [53]. The results revealed a significant reduction in goblet cell density (GCD) and goblet cell area (GCA) in DED mice compared to controls, further supporting the role of goblet cell loss in DED progression [53]. In our study, the number of conjunctival goblet cells (CGCs) was lower in the PHMG-p-treated group after DED induction compared to the DED group, except in the 0.3% PHMG-p only group. These findings confirm that PHMG-p exposure exacerbates DED. Notably, reduction in conjunctival goblet cell (CGC) was not as pronounced in the 0.3% PHMG-p only treated group compared to other groups is likely due to the strong cytotoxicity of high-dose PHMG-p, which may have suppressed immune cell activity. At higher concentrations, intense cytotoxic effects may have inhibited the immune response or induced early cell death, ultimately protecting goblet cells from immune-mediated damage. This finding suggests a concentration-dependent biphasic effect, where lower concentrations of PHMG-p may promote immune activation and inflammatory responses, while higher concentrations may lead to immune suppression and reduced inflammation, resulting in less goblet cell damage.

Additionally, several studies have established a strong association between DED and elevated levels of inflammatory cytokines, particularly IL-1β, IL-6, and TNF-α. These cytokines play a critical role in the pathogenesis and progression of DED, contributing to ocular surface inflammation and tear film instability. Research has consistently shown that IL-1β, IL-6, and TNF-α levels are significantly elevated in the tears and serum of DED patients compared to healthy controls [16,54,55]. Furthermore, a meta-analysis by Roda et al. (2020) confirmed that DED patients exhibit higher levels of these pro-inflammatory cytokines, reinforcing their role in the chronic inflammatory state characteristic of DED [55]. In our study, similar to previous studies, IL-1β, IL-6, and TNF-α levels tended to increase compared to the control group and were further elevated in the PHMG-p only group.

In our previous study, we identified biomarkers of DED through plasma and urinary metabolic profiling [45]. Plasma and urine metabolic profiling provide valuable insights into biochemical processes and have significant applications in research and clinical settings. One of the key advantages of urine metabolic profiling is its non-invasive nature, making sample collection more convenient and patient-friendly [56,57]. Additionally, both plasma and urine contain metabolic signatures from various biochemical pathways, offering a holistic representation of an individual’s metabolic status [58]. Another critical advantage of metabolic profiling is its role in biomarker discovery, enabling the identification of potential biomarkers for early disease detection, diagnosis, and prognosis [56,59,60]. Furthermore, urine provides a broader metabolite coverage as it contains end-products from multiple organs, reflecting systemic metabolic changes more comprehensively [56]. Also, urine samples exhibit high stability, making them ideal for metabolomic analysis [61]. By offering a comprehensive and quantitative assessment of metabolic processes, plasma and urine metabolic profiling serve as powerful tools for advancing our understanding of human health, disease mechanisms, and therapeutic monitoring.

In our study, clearer clustering observed in the plasma PCA profile compared to the urinary NMR profile likely reflects inherent differences in biological variability and metabolic dynamics between these two matrices. Plasma provides a more immediate snapshot of systemic metabolic changes, especially in response to acute toxicological stress, such as PHMG-p exposure. In contrast, urine reflects a time-averaged excretory profile that is influenced by hydration status, renal function, and other confounding physiological factors, which may contribute to relatively blurred clustering. Nevertheless, urine remains a valuable biological matrix in metabolomics due to its non-invasive nature, wide metabolite coverage, and ability to capture cumulative metabolic alterations. In our findings, although plasma offered clearer group separation, urinary analysis still yielded meaningful insights—particularly in identifying overlapping pathways such as glutamate, glycine, and citrate metabolism. These complementary features underscore the advantage of integrating both plasma and urine profiles to enhance the overall interpretation of systemic toxicity.

Our findings confirmed that many metabolic pathways were shared between plasma and urinary metabolic profiling, with the following pathways being consistent: alanine metabolism, amino sugar metabolism, ammonia recycling, arginine and proline metabolism, aspartate metabolism, carnitine synthesis, the citric acid cycle, fatty acid biosynthesis, gluconeogenesis, the glucose-alanine cycle, glutamate metabolism, glycine and serine metabolism, ketone body metabolism, nicotinate and nicotinamide metabolism, transfer of acetyl groups into mitochondria, and the urea cycle (Table 1 and Table 2). Among these, the metabolic pathways associated with DED include arginine and proline metabolism, the citric acid cycle, glutamate metabolism, and glycine and serine metabolism, among others, which will be discussed in detail below [42,47]. Notably, among the pathways related to dry eye disease, glutathione metabolism was the only one that differed between plasma and urine.

For the plasma and urinary metabolites arginine, glutamate, glutamine, glycine, and pyruvate, their involvement in pathways such as arginine and proline metabolism, glutamate metabolism, glycine and serine metabolism, and the urea cycle has been confirmed. These metabolites show a significant correlation with inflammatory cytokines, including TNF-α, IL-6, and IL-1β. In the current study, glutamine and glutamate was upregulated in PHMG-p treated group in both plasma or urine (Figure 6 and Figure 9). Glutamine plays a vital role in regulating inflammatory responses by supporting lymphocyte proliferation and cytokine production, making it essential for immune homeostasis. Studies have shown that glutamine supplementation reduces pro-inflammatory cytokine levels in the gut mucosa, including IL-1β, IL-6, and TNF-α, suggesting that glutamine availability can modulate immune activation and inflammatory pathways, contributing to a more controlled immune response [62]. In contrast, elevated extracellular glutamate levels have been linked to increased inflammation, particularly through neuroimmune interactions. Inflammatory processes can cause glutamate spillover into the extrasynaptic space, often as a result of glial dysfunction. This excess glutamate further exacerbates immune responses by activating microglial receptors, leading to the release of pro-inflammatory cytokines, such as TNF-α and IL-1β [63]. These findings highlight the dual role of glutamine and glutamate in inflammatory regulation, with glutamine acting as an anti-inflammatory agent, while excessive glutamate contributes to neuroinflammation and cytokine activation. Additionally, glutamate metabolism plays a critical role in inflammation and oxidative stress. Since glutamine can be converted to glutamate, it has been identified as an anti-inflammatory agent by inhibiting the production of reactive oxygen species (ROS), nitric oxide synthase (NOS), inducible nitric oxide synthase (iNOS), and cyclooxygenase-2 (COX-2), all of which are key mediators of inflammation [63]. Furthermore, glutamine suppresses NF-κB activation and inhibits the phosphorylation of STAT1, STAT5, and Akt, leading to reduced levels of TNF-α and IFN-γ [64]. However, at high concentrations, glutamate can induce oxidative stress and apoptosis in cerebral vascular endothelial cells, contributing to neuroinflammation and vascular damage [65]. These findings underscore the complex role of glutamate metabolism in inflammation, balancing both protective and detrimental effects depending on its concentration and metabolic context.

Glycine exhibits a significant anti-inflammatory effect by modulating the production and activity of pro-inflammatory cytokines, particularly TNF-α, IL-6, and IL-1β. Studies have demonstrated that glycine inhibits the production of these cytokines in various cell types and tissues, thereby contributing to the suppression of inflammatory responses [66,67]. This reduction in pro-inflammatory cytokine levels plays a crucial role in mitigating systemic inflammation, highlighting glycine’s potential as a therapeutic agent for inflammatory conditions. Beyond its anti-inflammatory properties, glycine is actively involved in glycine and serine metabolism, which is essential for immune regulation and cellular function. One of glycine’s key mechanisms in inflammation control is its ability to reduce calcium influx in macrophages, thereby inhibiting the production of toxic free radicals and ultimately reducing oxidative stress and tissue damage [64]. According to our study results, glycine levels were upregulated in all experimental groups compared to the control in plasma, and showed an increasing trend in urine, except in the PHMG-p 0.3% group (Figure 6 and Figure 9). This suggests that glycine levels may have increased in response to suppress the elevation of inflammatory cytokines TNF-α, IL-6, and IL-1β. These findings indicate that glycine and serine metabolism not only regulate inflammatory pathways but also contribute to disease progression in various pathological conditions, including immune-related disorders.

Pyruvate levels were up-regulated in the plasma after DED induction in the high-dose PHMG-p group and in the PHMG-p only group (Figure 6). Pyruvate levels typically increase during inflammation and oxidative stress, representing a metabolic adaptation to cellular stress. This elevation is associated with several protective mechanisms, as pyruvate and lactate contribute to oxidative stress resistance by inducing a mild hormetic increase in reactive oxygen species (ROS). This transient ROS surge activates antioxidant defenses and pro-survival pathways, ultimately enhancing cellular protection [68]. Additionally, pyruvate supplementation has been shown to suppress mitochondrial ROS generation and maintain mitochondrial membrane potential under oxidative stress conditions, highlighting mitochondria as key targets of pyruvate’s protective effects [69]. The rise in pyruvate levels during inflammation and oxidative stress may serve a protective function by modulating mitochondrial activity and maintaining redox balance, ultimately supporting cell survival and adaptation [68,69]. These findings suggest that pyruvate plays a critical role in cellular defense mechanisms against oxidative damage. Beyond its role in oxidative stress regulation, pyruvate is a key player in immune cell metabolism. Although not directly part of the TCA cycle, pyruvate serves as a crucial metabolic intermediary that influences immune cell activation and function. In activated immune cells, pyruvate is preferentially converted to lactate rather than entering the TCA cycle, a metabolic shift that supports immune cell activation and proliferation by facilitating glycolytic reprogramming [70]. However, a portion of pyruvate can still enter the TCA cycle to produce citrate, which is then exported to the cytosol to participate in immune-related functions, such as lipid biosynthesis and inflammatory signaling [71].

In our study, arginine levels were increased in plasma across all experimental groups (Figure 6). Arginine plays a crucial role in the urea cycle and nitric oxide (NO) production, making it an essential component in inflammation and oxidative stress regulation [64]. The urea cycle is closely linked to arginine metabolism, as it facilitates nitrogen excretion while also contributing to NO synthesis, which has significant implications for immune function and vascular homeostasis [64]. During inflammatory responses, macrophage M1 polarization promotes the expression of inducible nitric oxide synthase (iNOS), which utilizes arginine to generate NO. This process enhances the antimicrobial and immunomodulatory effects of macrophages while also inhibiting reactive oxygen species (ROS) activity, thereby influencing redox balance and inflammatory signaling [64]. Furthermore, arginine is synthesized from glutamine, glutamate, and proline via the intestinal-renal axis, highlighting its interconnected role in multiple metabolic pathways. Beyond its involvement in NO synthesis, arginine also contributes to creatine biosynthesis and methylation reactions, both of which are critical for maintaining cellular homeostasis and metabolic regulation [72].

The urea cycle is closely linked to inflammatory response, immune regulation, and oxidative stress, playing a critical role in maintaining metabolic and immune homeostasis. Disruptions in the urea cycle, such as urea cycle disorders (UCDs), can lead to hyperammonemia, which negatively affects immune cell function, particularly T cells and macrophages [73]. Additionally, elevated levels of certain amino acids, including glutamine, glycine, and alanine, in UCDs may influence T and B cell activity, further impacting immune function [73]. In our study, glutamine and glycine were increased in both plasma and urine, whereas alanine tended to increase only in plasma (Figure 6 and Figure 9). Beyond its metabolic role, the urea cycle is essential for proper immune responses, particularly in T-cell function. UCDs have been associated with significant immune dysfunctions, including impaired T-cell proliferation [73]. Arginine, a key intermediate in the urea cycle, is critical for T-cell activation and function, and its depletion can lead to increased susceptibility to infections [73].

The citric acid cycle (TCA cycle) plays a fundamental role in cellular metabolism and has significant implications for inflammation, immune responses, and oxidative stress regulation. Beyond its primary function in energy production, key metabolic intermediates such as citrate, succinate, and 2-oxoglutarate act as critical signaling molecules that influence immune cell function and inflammatory processes. In our study, the plasma metabolites associated with the citric acid cycle were citrate, pyruvate, and succinate, while the urinary metabolites included cis-aconitate, citrate, 2-oxoglutarate, and succinate. All plasma metabolites showed an increase, whereas in urine, citrate, 2-oxoglutarate, and succinate exhibited an increasing trend in the PHMG-p treatment groups (Figure 6 and Figure 9). Among these metabolites, citrate has emerged as a key immunometabolite, exerting multiple effects on inflammation and immune regulation. In pro-inflammatory macrophages, citrate accumulation promotes the production of inflammatory mediators, including prostaglandins, nitric oxide (NO), and reactive oxygen species (ROS), which further amplify immune responses [70,74]. Additionally, cytosolic citrate serves as a precursor for acetyl-CoA, which plays a crucial role in histone acetylation and epigenetic regulation, thereby influencing the expression of inflammatory genes [74]. Furthermore, citrate is essential for lipid biosynthesis, a process necessary for membrane expansion in macrophages and dendritic cells, which supports antigen presentation and cytokine production [71]. These findings highlight citrate’s dual role as both a metabolic substrate and a regulatory molecule in immune responses. Similarly, succinate accumulation in activated immune cells has profound effects on inflammatory signaling. One of its key functions is stabilizing hypoxia-inducible factor 1α (HIF-1α), which enhances the expression of pro-inflammatory genes, thereby promoting immune activation [75]. Additionally, succinate oxidation by succinate dehydrogenase (SDH) drives mitochondrial ROS production, further amplifying inflammation and immune cell activation [76]. These findings underscore succinate’s role as a key regulator of metabolic reprogramming in immune cells. 2-Oxoglutarate also plays a crucial role in immune cell regulation and metabolic sensing. It serves as a co-substrate for demethylases, influencing epigenetic modifications that govern immune cell differentiation and function [77]. Additionally, changes in the isocitrate-to-α-ketoglutarate ratio in pro-inflammatory macrophages serve as metabolic indicators of immune activation [70]. This suggests that 2-oxoglutarate is involved in both cellular adaptation to metabolic stress and the regulation of inflammatory responses. Overall, the Citric Acid Cycle and its intermediates play a critical role in orchestrating immune responses, inflammation, and oxidative stress regulation. These metabolites not only provide energy and biosynthetic precursors, but also function as signaling molecules that mediate metabolic reprogramming in immune cells, further influencing inflammatory and immune regulatory pathways.

Glutathione (GSH) metabolism, which occurs specifically in plasma, involves amino acids such as glycine, glutamate, and alanine and plays a crucial role in regulating inflammation, modulating immune responses, and protecting against oxidative stress. In our study, the levels of glycine, glutamate, and alanine tended to increase in the PHMG-p treatment group (Figure 6 and Figure 9). GSH metabolism plays a critical role in regulating inflammatory responses, immune function, and oxidative stress, serving as a key antioxidant and modulator of immune cell activity. The relationship between GSH and these processes is complex, as it influences cytokine production, cell survival, and redox balance. GSH plays an essential role in modulating pro-inflammatory cytokine production in response to pathogens. Intracellular GSH levels directly regulate the release of cytokines, and GSH depletion has been shown to partially suppress pro-inflammatory responses to certain stimuli, indicating its role in immune activation [78]. Additionally, GSH contributes to cell survival in monocytes exposed to inflammatory stimuli, highlighting its importance in maintaining immune cell function under inflammatory conditions [78]. These findings suggest that GSH is not only a protective antioxidant but also a key regulator of immune homeostasis. Beyond its role in inflammation, GSH is a major cellular antioxidant, crucial for buffering reactive oxygen species (ROS) generated during immune cell activation. By neutralizing ROS, GSH prevents oxidative damage to immune cells and other cellular components, ensuring proper immune function and cellular integrity [79]. These findings underscore the essential role of GSH in controlling oxidative stress and inflammation, making it a potential target for therapeutic interventions in inflammatory and oxidative stress-related diseases.

In conclusion, this study demonstrates that exposure to PHMG-p exacerbates DED by promoting inflammatory responses on the ocular surface. Metabolomic profiling identified distinct metabolic alterations in plasma and urine, particularly in pathways related to glutamate metabolism, glycine and serine metabolism, arginine and proline metabolism, glutathione metabolism, and the citric acid cycle. These metabolic changes correlated with elevated levels of pro-inflammatory cytokines (IL-1β, IL-6, and TNF-α) in corneal and lacrimal gland tissues, reinforcing the role of inflammation in PHMG-p-induced ocular toxicity. Furthermore, glutamate and glycine levels were significantly elevated in both plasma and urine, suggesting their involvement in immune regulation and oxidative stress. The observed increase in pyruvate, succinate, and citrate highlights the role of metabolic reprogramming in inflammatory processes. Notably, arginine levels were elevated in plasma, supporting its role in nitric oxide (NO) production and immune modulation. These findings underscore the potential risks of PHMG-p exposure on ocular health, particularly for individuals using humidifiers in environments prone to DED. Additionally, the metabolomic insights provided in this study contribute to a deeper understanding of DED pathophysiology, offering new perspectives for biomarker discovery and targeted therapeutic strategies.

## 4. Materials and Methods

### 4.1. Chemicals

Polyhexamethylene guanidine phosphate (PHMG-p, 25%) was obtained from SK Chemicals (Seoul, Republic of Korea). Scopolamine hydrobromide, 4,4-dimethyl-4-silapentane-1-sulfonic acid (DSS) and 3-(trimethylsilyl)-propionic-2,2,3,3-d4 acid sodium salt (TSP) were obtained from Sigma Aldrich (St. Louis, MO, USA). Sodium azide was purchased from Bio Basic Inc. (Markham, ON, Canada).

### 4.2. In Vivo Animal Study Design and Experimental Procedures

Forty-two male Sprague Dawley (S-D) rats (7 weeks old, weighing 200–240 g) were purchased from Samtako Co. (Osan, Republic of Korea). The experimental protocol was reviewed and approved by the Institutional Animal Care and Use Committee of Dankook University (IACUC approval number: 2017-035). All animals were acclimatized for one week under controlled environmental conditions: a 12-h light/dark cycle (lights on from 06:00 to 18:00), ambient temperature maintained at 20–24 °C, and relative humidity at 30% ± 5%. During the acclimatization and experimental periods, standard laboratory chow (LabDiet 5L79, Orientbio Inc., Seongnam, Republic of Korea) and tap water were provided ad libitum. The rats were randomly divided into seven groups (*n* = 6 per group). Group 1 served as the untreated control. Groups 2 to 5 were assigned as dry eye disease (DED) induction groups. DED was induced by subcutaneous (SC) injection of scopolamine hydrobromide at a dose of 3 mg/kg (total 12 mg/day), administered four times daily at 9:00, 12:00, 15:00, and 18:00 for 13 consecutive days. In addition to pharmacological induction, animals in these groups were exposed to desiccating environmental stress by housing them in a dry eye chamber for 18 h per day (from 16:00 to 10:00 the following day), followed by 6 h outside the chamber [80,81,82,83,84]. In the positive control (PC) group (Group 3), 0.1% benzalkonium chloride (BAC) in phosphate-buffered saline (PBS) was topically instilled into both eyes once daily at 15:00, with a total of 12 applications per session at 5-min intervals, in addition to scopolamine administration and chamber exposure. Groups 4 and 5 followed the same scopolamine and desiccation regimen as Group 2, but instead received topical PHMG-p at concentrations of 0.1% and 0.3%, respectively, applied using the same method and schedule as BAC in Group 3. Groups 6 and 7 were treated with PHMG-p alone at 0.1% and 0.3%, respectively, following the same topical application schedule and environmental exposure as the other groups. To confirm successful DED induction, diagnostic assessments were conducted on day 8 of scopolamine administration. These included measurement of tear volume, tear film break-up time (TBUT), and microscopic examination of the ocular surface. On day 14, after completion of all treatments, the same diagnostic parameters were evaluated to assess the therapeutic or aggravating effects of the interventions. Final assessments included tear volume, corneal fluorescein staining scores, and TBUT. At the end of the experiment, rats were euthanized for tissue and biofluid collection. Eyeballs were carefully removed for histological evaluation. Urine samples were collected overnight in metabolic cages following the final scopolamine administration. Each urine sample was collected into a glass bottle containing 50 µL of 3% sodium azide. Blood samples were obtained via the abdominal aorta under CO_2_ anesthesia using an 18-gauge needle, and centrifuged at 3000× *g* for 15 min at 4 °C to separate plasma. All urine and plasma samples were stored at −70 °C until further analysis.

### 4.3. Clinical Evaluation of Animal Models

Tear volume was measured using phenol red-impregnated cotton threads (Zone-Quick, Glendora, CA, USA). The threads were gently placed at the lateral canthus of each eye using fine forceps and removed after 20 s. The wetted length of the thread, which turned red upon contact with tears, was measured under a microscope and expressed in mm [82]. The measured length was converted to tear volume based on the manufacturer’s instructions. Tear break-up time (TBUT) was assessed to evaluate tear film stability. A 1% sodium fluorescein solution was instilled into both eyes of non-anesthetized rats, and the time until the appearance of dry spots on the corneal surface after blinking was measured under cobalt blue light. The average of three repeated measurements was recorded as the TBUT. Corneal fluorescein staining was performed to assess epithelial damage. A 5% fluorescein solution was applied to the conjunctival sac, and eyes were examined under a cobalt blue light using a slit-lamp biomicroscope. Corneal damage was scored from 0 to 4 according to the severity of clinical signs, including conjunctival congestion, secretions, conjunctival edema, or edema of the eyelids.

### 4.4. Histological and Histochemical Analysis

Hematoxylin and eosin (H&E) staining was performed for general morphological evaluation. Whole eyes were surgically excised, fixed in 4% formalin, and embedded in optimal cutting temperature (O.C.T.) compound (Tissue-Tek, Sakura, Tokyo, Japan). Corneal and conjunctival tissues were cryosectioned at a thickness of 5 μm using a cryomicrotome, and stained with H&E. Stained sections were imaged using a virtual microscope system. Periodic acid–Schiff (PAS) staining was conducted to detect goblet cell. Cryosectioned tissues (5 μm) were treated with 0.5% periodic acid (Sigma–Aldrich, Inc, St. Louis, MO, USA, PAS kit.) for 5 min, rinsed with distilled water, and incubated with Schiff’s reagent for 30 min at room temperature. After washing under running water for 5 min, the sections were counterstained with hematoxylin for 3 min, dehydrated, cleared, and mounted. Digital images of the stained tissues were obtained using a virtual microscope.

### 4.5. Immunohistochemistry (IHC)

Tissue slides were fixed in acetone for 15 min and then blocked with 5% bovine serum albumin (BSA) (Albumin, Bioshop Canada Inc., Burlington, ON, Canada) in PBS for 1 h at room temperature. Primary antibodies against IL-1β (Abcam, Cambridge, UK; ab9722), IL-6 (AbPRONTIER, Seoul, Republic of Korea; YF-MA10477), and TNF-α (Abcam, Cambridge, UK; ab199013), each diluted 1:200 in 5% BSA, were applied and incubated overnight at 4 °C. After three washes in PBS (5 min each), enzyme-conjugated secondary antibodies diluted in 5% BSA were applied for 1 h at room temperature. Following another round of PBS washing, 2.5 mL of 5% BSA containing reagents A and B from the ABC kit (VECTASTAIN Elite ABC HRP Kit, Vector Laboratories, Inc., Burlingame, CA, USA; Cat. No: PK-6100) was added and incubated for 30 min. After washing with PBS for 5 min, a drop of buffer stock solution, a drop of 3,3′-diaminobenzidine (DAB) stock buffer, and 25 μL hydrogen peroxide solution of the DAB Substrate Kit (DAB Peroxidase Substrate, Vector Laboratories, Inc., 30 Ingold Road, Burlingame, CA, USA; Cat. No: SK-4100) were added to 2.5 mL 5% BSA in PBS until a browning reaction appeared. The sections were counterstained with Mayer’s hematoxylin (Sigma-Aldrich, Inc, St. Louis, MO, USA) for 30 s and imaged using a virtual microscope.

### 4.6. ^1^H NMR-Based Metabolomics Analysis

After thawing the plasma samples at 4 °C, a 350 μL aliquot was transferred into a microcentrifuge tube containing 350 μL of deuterated water (D_2_O) solution with 4 mM TSP as an internal standard for chemical shift referencing. Urine samples were also thawed at 4 °C and centrifuged to remove particulates. A 600 μL aliquot of the urine supernatant was added to a microcentrifuge tube containing 70 μL of D_2_O solution supplemented with 5 mM DSS and 100 mM imidazole. In addition, 30 μL of 0.42% sodium azide was added to each urine sample to prevent microbial growth. Following vortex mixing, all samples were subjected to ^1^H-NMR analysis within 48 h. NMR spectra were acquired using a Varian Unity Inova 600 MHz spectrometer (Agilent Technologies, Santa Clara, CA, USA) operating at 26°C, located at Pusan National University (Busan, Republic of Korea). The Carr–Purcell–Meiboom–Gill (CPMG) pulse sequence was applied to attenuate signals from macromolecules and residual water. For urine samples, spectra were acquired using the following parameters: a 16.2 μs 90° pulse width, a 3 s relaxation delay, a 3 s acquisition time, and a total acquisition time of 13 min and 9 s. For plasma samples, the acquisition was performed using a 16.5 μs 90° pulse width, a 3 s relaxation delay, a 3 s acquisition time, and a total acquisition time of 13 min and 20 s. All spectra were acquired with 128 scans and a spectral width of 24,038.5 Hz. Spectral data were processed using the Chenomx NMR Suite software (version 8.3, Chenomx Inc., Edmonton, AB, Canada). The spectral region from δ 0.0 to 10.0 ppm was divided into bins with a width of 0.04 ppm, yielding a total of 250 integrated regions per spectrum. This binning procedure generated a normalized intensity distribution across 250 variables for each spectrum, which was used for subsequent pattern recognition analysis. The water resonance region (δ 4.5–5.0 ppm) was excluded from the analysis to minimize variability associated with differences in water suppression efficiency. Metabolite identification and quantification were also performed using the Chenomx NMR Suite Professional software (version 8.3). TSP (2 mM) and DSS (0.5 mM) were used as the chemical shift and concentration references for plasma and urine samples, respectively.

### 4.7. Multivariate and Statistical Analysis

All NMR spectral data were exported from the Chenomx NMR Suite Professional software into Microsoft Excel (*.xls) format. The one-dimensional NMR data were then imported into SIMCA-P software (version 12.0, Umetrics Inc., Kinnelon, NJ, USA) for multivariate statistical analysis to explore intrinsic variations in the dataset. Prior to analysis, the data were center-scaled. PCA and OPLS-DA were performed, and the resulting score plots were used to visualize the distribution and clustering of samples across groups. Additionally, VIP values from the OPLS-DA model were used to identify potential biomarkers associated with PHMG-p exposure in the DED model. For univariate analysis, the means and standard deviations of each metabolite were calculated using Microsoft Excel (version 2019). Quantitative differences in plasma and urinary metabolite concentrations among experimental groups were assessed using ANOVA test in GraphPad Prism (version 5.01, San Diego, CA, USA). A *p*-value of < 0.05 was considered statistically significant.

## Figures and Tables

**Figure 1 ijms-26-08660-f001:**
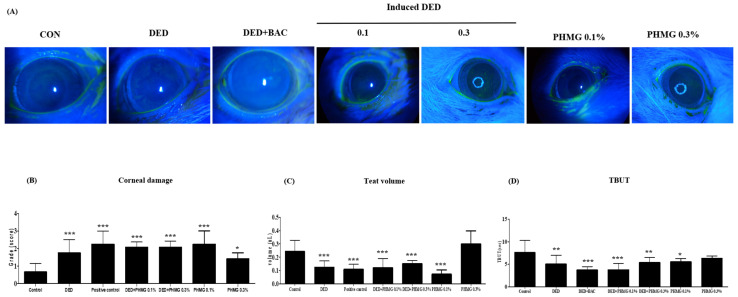
Images of the eyes of rat in the control groups, DED induced group, positive control group and PHMG-p treated groups were photographed with a microscope after 13 day desiccation stress and after the instillation of scopolamine eye drops. (**A**) Corneal fluorescein staining. (**B**) Corneal damage. (**C**) Tear volume. (**D**) Tear break-up time (TBUT). * *p* < 0.05, ** *p* < 0.01, *** *p* < 0.001 compared with the control group.

**Figure 2 ijms-26-08660-f002:**
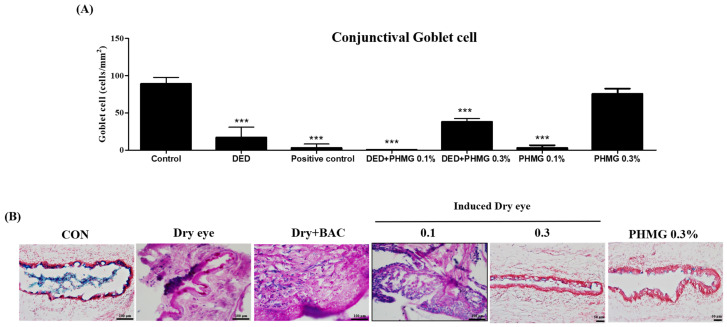
Effect of control groups, DED induced group, positive control group and PHMG-p treated groups on detached conjunctive goblet cell of the rats. (**A**) Quantitative analysis of conjunctive goblet cell density (cells/mm^2^) in each group. (**B**) Representative histological image of conjunctive goblet cell. The result of each group are shown as mean ± SD. *** *p* < 0.001 compared with the control group.

**Figure 3 ijms-26-08660-f003:**
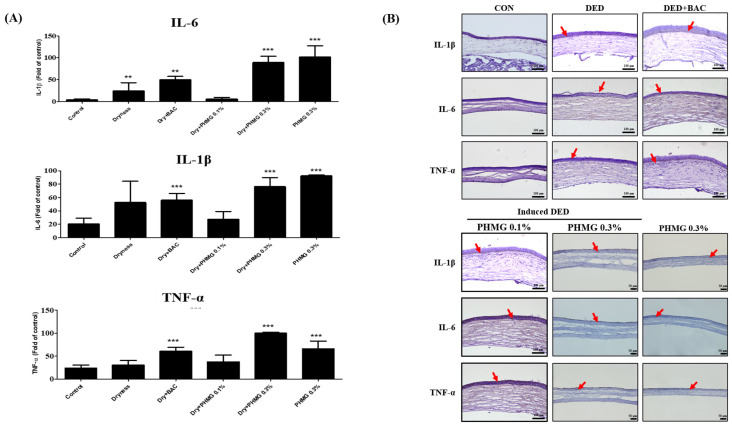
The change of immune index markers in the cornea epithelium of the rats. (**A**) Quantitative analysis of IL-6, IL-1β, and TNF-α expression in the cornea epithelium. (**B**) Representative immunohistochemical images. Values are shown as mean ± SD. ** *p* < 0.01, *** *p* < 0.001 compared with the control group. Red arrows indicate cytokine expression.

**Figure 4 ijms-26-08660-f004:**
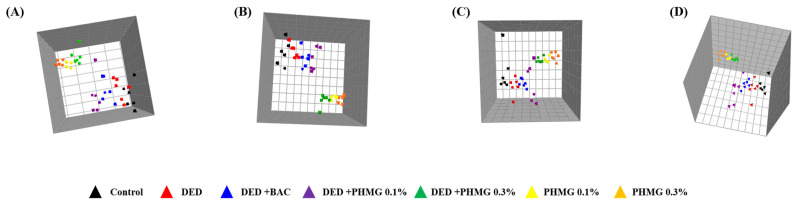
In global profiling, principal component analysis (PCA) (R^2^X = 0.749, Q^2^ = 0.666) models (**A**) and orthogonal projections to latent structures-discriminant analysis (OPLS-DA) (R^2^X = 0.738, R^2^Y = 0.181, Q^2^ = 0.0871) models (**B**) results after NMR analysis of control, DED group and PHMG-p treated groups in plasma samples. Global profiling of PCA (R^2^X = 0.883, Q^2^ = 0.825) (**C**) and OPLS-DA (R^2^X = 0.882, R^2^Y = 0.174, Q^2^ = 0.0955) (**D**) using NMR data from urine samples. ▲, Control; ▲, DED group; ▲, DED + BAC; ▲, DED + PHMG-p 0.1%; ▲, DED + PHMG-p 0.3%; ▲, PHMG-p 0.1%; ▲, PHMG-p 0.3%.

**Figure 5 ijms-26-08660-f005:**
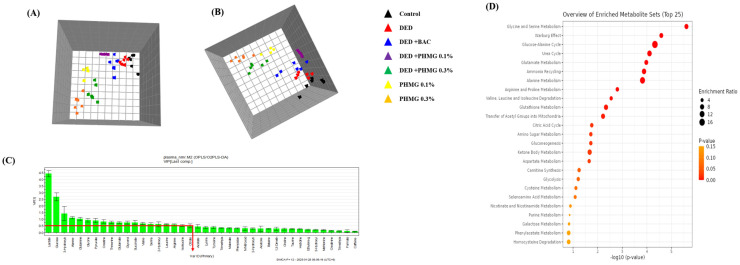
In target profiling, principal component analysis (PCA) (R^2^X = 0.947, Q^2^ = 0.816) models (**A**) and orthogonal projections to latent structures-discriminant analysis (OPLS-DA) (R^2^X = 0.991, R^2^Y = 0.585, Q^2^ = 0.109) models (**B**) results after NMR analysis of control and DED group in plasma sample. Variable importance plot (VIP) (**C**) shows the major urine metabolites that contributed to separate the clusters. Metabolites set enrichment overview in plasma samples (**D**). ▲, Control; ▲, DED group; ▲, DED + BAC; ▲, DED + PHMG-p 0.1%; ▲, DED + PHMG-p 0.3%; ▲, PHMG-p 0.1%; ▲, PHMG-p 0.3%.

**Figure 6 ijms-26-08660-f006:**
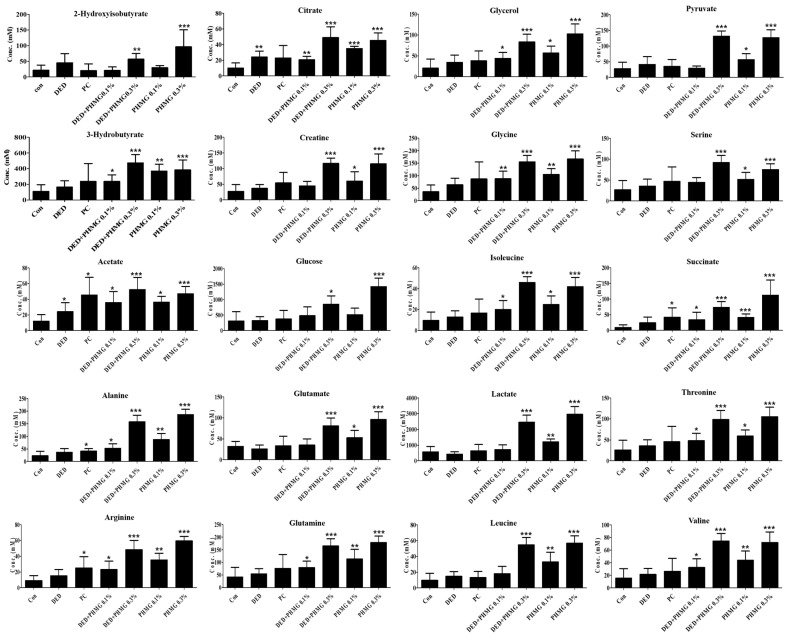
Concentrations of endogenous metabolites in plasma samples for DED induced and PHMG-p treatment to rats. ANOVA test was performed to assess statistical significance compared with control and treatment. Error bars are expressed as S.D. * *p* < 0.05, ** *p* < 0.01, *** *p* < 0.001.

**Figure 7 ijms-26-08660-f007:**
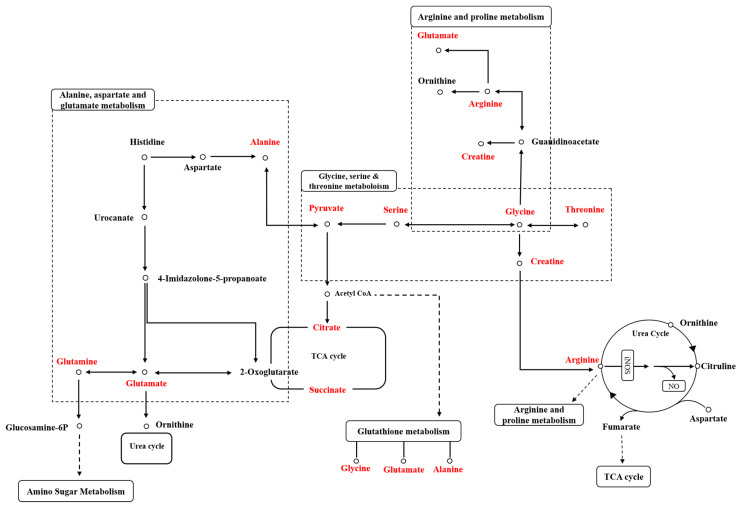
Expected plasma metabolic pathway network of metabolites related to PHMG-p exposure.

**Figure 8 ijms-26-08660-f008:**
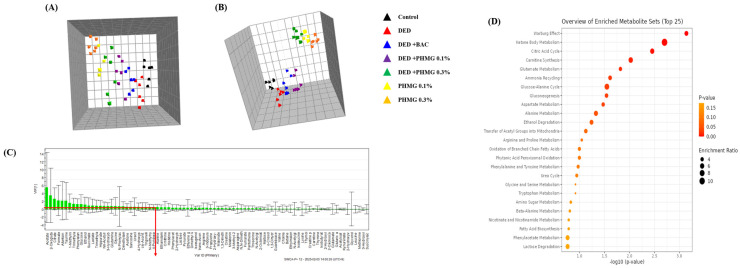
Principal component analysis (PCA) (R^2^X = 0.463, Q^2^ = 0.0756) (**A**) model and orthogonal projections to latent structures-discriminant analysis (OPLS-DA) (R^2^X = 0.832, R^2^Y = 0.586, Q^2^ = 0.31) (**B**) in target profiling of urine samples. Variable importance plot (VIP) (**C**) shows the major urine metabolites that contributed to separate the clusters. Metabolites set enrichment overview in urine samples (**D**). ▲, Control; ▲, DED group; ▲, DED+BAC; ▲, DED+PHMG-p 0.1%; ▲, DED+PHMG-p 0.3%; ▲, PHMG-p 0.1%; ▲, PHMG-p 0.3%.

**Figure 9 ijms-26-08660-f009:**
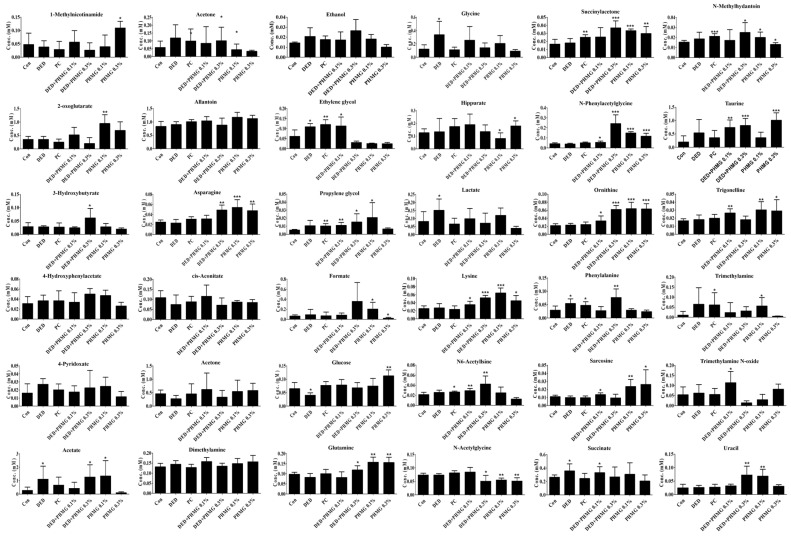
Concentrations of endogenous metabolites in urine samples for DED induced and PHMG-p treatment to rats. ANOVA test was performed to assess statistical significance compared with control and treatment. Error bars are expressed as S.D. * *p* < 0.05, ** *p* < 0.01, *** *p* < 0.001.

**Figure 10 ijms-26-08660-f010:**
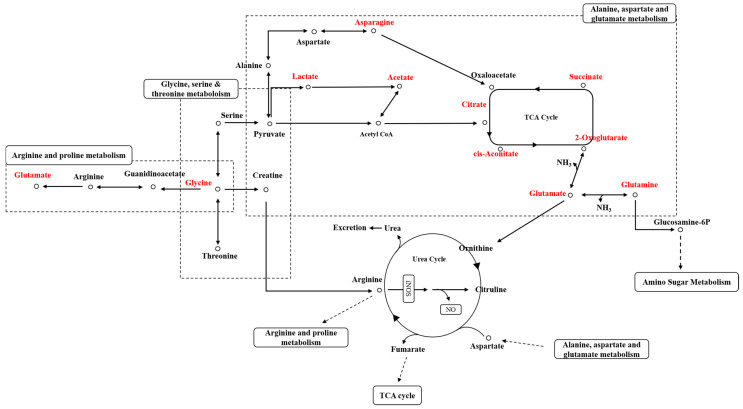
Expected urinary metabolic pathway network of metabolites related to PHMG-p exposure.

**Table 1 ijms-26-08660-t001:** Expected pathway based on high-VIP-scored (VIP > 0.5) metabolites in rat plasma using MetaboAnalyst (6.0).

Plasma							
Pathway	Total	Expected	Hits	*p* Value	−log(*p*)	Holm Adjust	FDR
Glycine and Serine Metabolism	59	1.06	8	2.48 × 10^−6^	0.000243	0.000243	0.000243
Glucose-Alanine Cycle	13	0.234	4	4.73 × 10^−5^	0.00454	0.00155	0.00155
Urea Cycle	28	0.503	5	7.85 × 10^−5^	0.00746	0.00192	0.00192
Glutamate Metabolism	48	0.862	6	0.000106	0.00994	0.00207	0.00207
Ammonia Recycling	31	0.557	5	0.000131	0.0122	0.00211	0.00211
Alanine Metabolism	17	0.305	4	0.000151	0.0139	0.00211	0.00211
Arginine and Proline Metabolism	52	0.934	5	0.00159	0.145	0.0195	0.0195
Valine, Leucine and Isoleucine Degradation	59	1.06	5	0.00283	0.255	0.0308	0.0308
Glutathione Metabolism	20	0.359	3	0.00459	0.409	0.045	0.045
Transfer of Acetyl Groups into Mitochondria	22	0.395	3	0.00606	0.533	0.054	0.054
Citric Acid Cycle	32	0.575	3	0.0174	1	0.133	0.133
Amino Sugar Metabolism	33	0.593	3	0.019	1	0.133	0.133
Gluconeogenesis	33	0.593	3	0.019	1	0.133	0.133
Ketone Body Metabolism	13	0.234	2	0.0212	1	0.136	0.136
Aspartate Metabolism	35	0.629	3	0.0222	1	0.136	0.136
Carnitine Synthesis	22	0.395	2	0.0569	1	0.328	0.328
Glycolysis	23	0.413	2	0.0617	1	0.336	0.336
Cysteine Metabolism	26	0.467	2	0.0768	1	0.396	0.396
Selenoamino Acid Metabolism	27	0.485	2	0.0821	1	0.402	0.402
Nicotinate and Nicotinamide Metabolism	35	0.629	2	0.128	1	0.544	0.544
Fatty Acid Biosynthesis	35	0.629	1	0.476	1	0.992	0.992

**Table 2 ijms-26-08660-t002:** Expected pathway based on high-VIP-scored (VIP > 0.5) metabolites in rat urine using MetaboAnalyst (6.0).

Urine							
Pathway	Total	Expected	Hits	*p* Value	−log(*p*)	Holm Adjust	FDR
Ketone Body Metabolism	13	0.272	3	0.00199	0.193	0.0973	0.0973
Citric Acid Cycle	32	0.671	4	0.00351	0.337	0.115	0.115
Carnitine Synthesis	22	0.461	3	0.00946	0.899	0.232	0.232
Glutamate Metabolism	48	1.01	4	0.0152	1	0.298	0.298
Ammonia Recycling	31	0.65	3	0.0245	1	0.354	0.354
Glucose-Alanine Cycle	13	0.272	2	0.0284	1	0.354	0.354
Gluconeogenesis	33	0.692	3	0.0289	1	0.354	0.354
Aspartate Metabolism	35	0.734	3	0.0337	1	0.367	0.367
Alanine Metabolism	17	0.356	2	0.0471	1	0.462	0.462
Ethanol Degradation	19	0.398	2	0.0577	1	0.514	0.514
Transfer of Acetyl Groups into Mitochondria	22	0.461	2	0.0751	1	0.614	0.614
Arginine and Proline Metabolism	52	1.09	3	0.0906	1	0.626	0.626
Oxidation of Branched Chain Fatty Acids	26	0.545	2	0.101	1	0.626	0.626
Phytanic Acid Peroxisomal Oxidation	26	0.545	2	0.101	1	0.626	0.626
Phenylalanine and Tyrosine Metabolism	27	0.566	2	0.107	1	0.626	0.626
Urea Cycle	28	0.587	2	0.114	1	0.626	0.626
Glycine and Serine Metabolism	59	1.24	3	0.121	1	0.626	0.626
Tryptophan Metabolism	59	1.24	3	0.121	1	0.626	0.626
Amino Sugar Metabolism	33	0.692	2	0.15	1	0.683	0.683
Beta-Alanine Metabolism	34	0.713	2	0.157	1	0.683	0.683
Nicotinate and Nicotinamide Metabolism	35	0.734	2	0.165	1	0.683	0.683
Fatty Acid Biosynthesis	35	0.734	2	0.165	1	0.683	0.683
Phenylacetate Metabolism	9	0.189	1	0.174	1	0.683	0.683
Lactose Degradation	9	0.189	1	0.174	1	0.683	0.683
Malate-Aspartate Shuttle	10	0.21	1	0.192	1	0.722	0.722
Taurine and Hypotaurine Metabolism	12	0.251	1	0.226	1	0.762	0.762

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
