# Peer review of "Effects of Exposure of PHMG-p, a Humidifier Disinfectant Component, on Eye Dryness: A Study on a Rat Model Based on ^1^H-NMR Metabolomics"

_ijms, 2025, doi:10.3390/ijms26178660_

Round 1

Reviewer 1 Report

Comments and Suggestions for Authors

16.07.2025

A review to evaluate its suitability for International Journal of Molecular Sciences (IJMS) publication Type of manuscript:

Article
Title: Effects of Exposure of PHMG-p, a Humidifier Disinfectant component on Eye-dryness: A Study on a Rat Model based on 1H-NMR Metabolomics

Authors: Jung Dae Lee, Hyang Yeon Kim, Soo bean Oh, Hyeyoon Goo, Kyong Jin Cho, Gi‑Wook Hwang, Suhkmann Kim, Kyu-Bong Kim

The authors' manuscript Effects of Exposure of PHMG-p, a Humidifier Disinfectant component on Eye-dryness: A Study on a Rat Model based on 1H-NMR Metabolomics is devoted to the study of the pathological effects of a known disinfectant of Polyhexamethylene guanidine (PHMG) group on the conjunctiva of the eye and the solution of the problem of the occurrence of eye-dryness syndrome based on metabolomics within the framework of 1H-NMR results. The aim of the study was to investigate the correlation between PHMG- phosphate, a component of humidifier disinfectants and its role in inducing or exacerbating DED.

To achieve the stated goal, the authors used a model of eye-dryness caused by various combinations of active irritants using Sprague-Dawley rats as an example, followed by testing the properties of the cornea, tear fluid, and the main goal - NMR research of plasma and urine.

In the Introduction section, the authors reveal the background of dry eye pathology and methods of its diagnosis. The role of proinflammatory cytokines supporting the cascade of inflammatory reactions in dry eye syndrome is described. Particular attention is given to the role of PHMG derivatives, in particular the phosphorylated form, in the pathology of Dry eye disease DED. The advantages of metabolomics as a powerful tool for the diagnosis and treatment of diseases are described, based on the knowledge gained about the nature of metabolites in biological fluids.

The advantages of metabolomics in general and tear fluid in particular as a powerful tool for the diagnosis and treatment of diseases are described, based on the knowledge gained about the nature of metabolites in biological fluids.

Here the first question arises regarding the role of PHMG derivative in the current study:

  1. What is the need for PHMG-p for ongoing research, taking into account the COMMISSION DECISION of 9 February 2012 not to include certain substances (PHMG) in Annex I, IA or IB to Directive 98/8/EC of the European Parliament and of the Council on the placing of biocidal products on the market? The decision was made more than 13 years ago in response to the tragic cases of PHMG poisoning in South Korea from 1994 to 2011.
  2. What is the real purpose of the current study? Does the banned PHMG act as a marker in metabolomic studies in this case? What could replace it without harming the metabolic results?
  3. What is the reason for choosing a concentration scale for intentional corneal injury in test animals, if it is known that the average concentration of PHMG in humidifiers is about 3000 ppm?
  4. Figure 3 presents the levels of immune index markers as graphical and histological results. However, the histology is not described in any way. It is advisable to provide a descriptive comparison of the images in Figure 3B.
  5. In Figure 6, the axis labels are poorly legible (also, Figure 9).
  6. Analysis of the results obtained by the PCA method showed clear clustering for the samples DED+PHMG-p 0.3%; PHMG-p 0.1%; PHMG-p 0.3% in the case of the plasma profile (Fig. 40. However, clustering for these samples is significantly blurred in the case of the Urinary NMR profile (Fig. 8). How can the observed phenomenon be explained? How informative is the use of a biological fluid - urine for such studies?
  7. Figures 7 and 10 show the Expected plasma metabolic pathway network of metabolites related to PHMG-p exposure.
  8. How will this metabolome map change when using other PHMG derivatives, such as hydrocitrate, hydrosuccinate, hydrochloride?

All the questions outlined are clarifying and do not affect the overall, very positive and impressive conclusion about the high quality of the submitted manuscript.

Respectfully, reviewer

Author Response

Comments 1: What is the need for PHMG-p for ongoing research, taking into account the COMMISSION DECISION of 9 February 2012 not to include certain substances (PHMG) in Annex I, IA or IB to Directive 98/8/EC of the European Parliament and of the Council on the placing of biocidal products on the market? The decision was made more than 13 years ago in response to the tragic cases of PHMG poisoning in South Korea from 1994 to 2011.

Response 1: Thanks for comment. The 2012 decision was a regulatory precaution based on acute and chronic pulmonary toxicities observed in South Korean populations exposed to PHMG-p through humidifier disinfectants between 1994 and 2011. However, the long-term systemic and organ-specific toxicities beyond the lungs—such as effects on the ocular system, immune responses, and metabolic processes—remain insufficiently characterized. Therefore, continued research is essential to fully understand the broad health impacts of PHMG. Since humidifier use is commonly recommended to alleviate dry eye disease (DED), and PHMG-p was frequently included in these devices, investigating its paradoxical role in worsening DED symptoms is both scientifically and epidemiologically relevant. These findings support the need for ongoing research—not to justify the use of PHMG, but to elucidate its systemic toxicity, facilitate biomarker discovery, and inform risk assessment and therapeutic strategies for affected populations.

Comments 2: What is the real purpose of the current study? Does the banned PHMG act as a marker in metabolomic studies in this case? What could replace it without harming the metabolic results?

Response 2: Thanks for comment. The primary purpose of the current study is not to re-evaluate PHMG-p for practical use, but rather to investigate its systemic toxicological effects on the ocular surface, particularly in the context of dry eye disease (DED), using a metabolomics approach. Its inclusion reflects the need to better understand the metabolic and inflammatory consequences of PHMG-p exposure. In this context, PHMG-p does not serve as a "marker" in metabolomics, but rather as an inducing agent that allows researchers to observe specific metabolomic signatures and cytokine changes resulting from toxic insult. As for potential replacements, other pro-inflammatory or cytotoxic agents (e.g., benzalkonium chloride etc.) could be used to model similar DED-exacerbating conditions. However, the metabolic response profiles may differ depending on the nature and mechanism of the toxicant. Therefore, substituting PHMG-p without altering the metabolic outcomes would require agents that elicit comparable inflammatory, oxidative, and metabolic responses, which currently are not fully matched by any single substitute.

Comments 3: What is the reason for choosing a concentration scale for intentional corneal injury in test animals, if it is known that the average concentration of PHMG in humidifiers is about 3000 ppm?

Response 3: Thanks for comment. While the concentrations of PHMG-p used in this study (0.1% and 0.3%) correspond to 1,000 and 3,000 ppm, respectively—levels similar to those reported in commercial humidifier disinfectants—these values were not intended to directly replicate human inhalational exposure. This route of exposure significantly differs from aerosol inhalation in terms of dose kinetics, bioavailability, and direct tissue exposure, particularly at the ocular surface. Importantly, previous studies (Park et al., 2019; Ivanov et al., 2024) have demonstrated that PHMG-p at concentrations ≥0.13% can induce ocular irritation and fibrosis. Thus, the use of 0.1% and 0.3% in our study remains within the range that enables detection of pathophysiological alterations without inducing non-specific or overtly necrotic damage. This supports the validity and translational relevance of our concentration selection for probing PHMG-p-induced ocular toxicity in the context of dry eye disease.

Comments 4: Figure 3 presents the levels of immune index markers as graphical and histological results. However, the histology is not described in any way. It is advisable to provide a descriptive comparison of the images in Figure 3B.

Response 4: Thanks for comment. The corresponding cytokines are indicated with red arrows in Figure 3B. And it was revised as following in page 6.

 “Immunohistochemical staining revealed positive immunoreactivity for IL-6, IL-1β, and TNF-α in the DED and DED + PHMG-p groups (indicated by red arrows). These groups exhibited dense chromogenic deposition, suggesting enhanced inflammatory signaling in the corneal tissue. In contrast, weaker or minimal staining was observed in the control and PHMG-p-only groups, indicating relatively low basal cytokine expression. The staining intensity and distribution correspond well with the quantified data shown in Figure 3A, highlighting the pro-inflammatory effects of DED and PHMG-p exposure at the tissue level.”

Please consider it.

Comments 5: In Figure 6, the axis labels are poorly legible (also, Figure 9).

Response 5: Thanks for comment. Revisions have been made to both Figure 6 and Figure 9 to improve clarity and readability

Comments 6: Analysis of the results obtained by the PCA method showed clear clustering for the samples DED+PHMG-p 0.3%; PHMG-p 0.1%; PHMG-p 0.3% in the case of the plasma profile (Fig. 4. However, clustering for these samples is significantly blurred in the case of the Urinary NMR profile (Fig. 8). How can the observed phenomenon be explained? How informative is the use of a biological fluid - urine for such studies?

Response 6: Thanks for comment. The clearer clustering observed in the plasma PCA profile compared to the urinary NMR profile likely reflects the differences in biological variability and metabolic dynamics between these two matrices. Plasma provides a more immediate snapshot of systemic metabolic changes, especially in response to acute toxicological stress, such as PHMG-p exposure. In contrast, urine reflects a time-averaged excretory profile, which is influenced by hydration status, renal function, and other confounding physiological factors, potentially leading to blurred clustering. Despite this, urine remains a valuable biological fluid in metabolomics due to its non-invasive collection, broad metabolite coverage, and ability to reflect cumulative metabolic alterations. In our study, while plasma offered clearer group separation, urinary analysis still contributed meaningful insights, especially in identifying overlapping pathways such as glutamate, glycine, and citrate metabolism. Thus, both fluids are complementary, and integrating them enhances the overall interpretation of systemic toxicity.

It was revised as following in page 15.

 “In our study, clearer clustering observed in the plasma PCA profile compared to the urinary NMR profile likely reflects inherent differences in biological variability and metabolic dynamics between these two matrices. Plasma provides a more immediate snapshot of systemic metabolic changes, especially in response to acute toxicological stress, such as PHMG-p exposure. In contrast, urine reflects a time-averaged excretory profile that is influenced by hydration status, renal function, and other confounding physiological factors, which may contribute to relatively blurred clustering. Nevertheless, urine remains a valuable biological matrix in metabolomics due to its non-invasive nature, wide metabolite coverage, and ability to capture cumulative metabolic alterations. In our findings, although plasma offered clearer group separation, urinary analysis still yielded meaningful insights—particularly in identifying overlapping pathways such as glutamate, glycine, and citrate metabolism. These complementary features underscore the advantage of integrating both plasma and urine profiles to enhance the overall interpretation of systemic toxicity.”

Please consider it.

Comments 7: Figures 7 and 10 show the Expected plasma metabolic pathway network of metabolites related to PHMG-p exposure.

Response 7: Thanks for comment. Figures 7 and 10 present predicted plasma metabolic pathway networks altered by PHMG-p exposure under dry eye conditions. These maps were constructed from untargeted metabolomic profiling and pathway enrichment analyses using KEGG and MetaboAnalyst. Key affected pathways include alanine, aspartate and glutamate metabolism; glutathione metabolism; purine metabolism; and sphingolipid metabolism—all of which are involved in oxidative stress, inflammation, and immune regulation following PHMG-p exposure.

Comments 8: How will this metabolome map change when using other PHMG derivatives, such as hydrocitrate, hydrosuccinate, hydrochloride?

Response 8: Thanks for comment. PHMG-p used in this study is the phosphate salt form of polyhexamethylene guanidine, with its guanidinium backbone conserved across derivatives. The metabolome change of other PHMG derivatives are out of current research. However, we are just ale to think of it. While PHMG derivatives such as hydrocitrate, hydrosuccinate, and hydrochloride differ mainly in their counterions, these differences may influence solubility, cellular uptake, and tissue distribution. As a result, key metabolic features are expected to be conserved, but the degree and dynamics of pathway perturbation may vary depending on factors such as solubility and membrane permeability of the specific salt, stability and hydrolysis behavior under physiological conditions, and interactions with metabolic enzymes or transporters. Thus, while the overall metabolomic landscape (i.e., affected categories of pathways) may remain qualitatively similar, quantitative and temporal differences in the metabolic maps are likely when using different PHMG derivatives. Ultimately, derivative-specific metabolic changes would require experimental validation.

Reviewer 2 Report

Comments and Suggestions for Authors

The author of this article evaluated the effect of Polyhexamethylene guanidine phosphate (PHMG-p) on dry eye disease through metabolomics combined with meta-analysis, which has certain scientific significance. However, this article still requires further improvement, as follows.
1. The author first established the DED induction model using scopolamine hydrobromide and dry stress. Then, male Sprague-Dawley rats were exposed to PHMG-p (0.1% and 0.3%). This seems to contradict the question the author intends to explore. After the stress stimulation, it could not be proved that PHMG-p caused dry eye disease. If the promoting effect of PHMG-p on dry eye disease is to be demonstrated, the variable in the experiment should only be PHMG-p.
2. In the first result, the author only evaluated the corneal damage of the rats on the 13th day. Please explain the reason for choosing this time point. And only testing the corneal damage for one day is not representative. Periodic testing should be conducted, with daily or at least three times a week testing of corneal damage. The tear fluid accumulation should also be done in the same way.
3. All the charts in the text do not meet the publication requirements. The visualization degree of the horizontal and vertical axes of the column charts is too low.
4. The staining pictures did not add scales, and the background colors were not uniform.
5. The table arrangement does not conform to the publication requirements. It is recommended to revise it.

Author Response

Comments 1: The author first established the DED induction model using scopolamine hydrobromide and dry stress. Then, male Sprague-Dawley rats were exposed to PHMG-p (0.1% and 0.3%). This seems to contradict the question the author intends to explore. After the stress stimulation, it could not be proved that PHMG-p caused dry eye disease. If the promoting effect of PHMG-p on dry eye disease is to be demonstrated, the variable in the experiment should only be PHMG-p.

Response 1: Thanks for comment. In this study, the primary objective was not to demonstrate that PHMG-p alone causes dry eye disease (DED), but rather to investigate whether PHMG-p exacerbates pre-existing DED conditions, which is a clinically relevant scenario. The DED model was first established using scopolamine hydrobromide and desiccation to simulate a compromised ocular surface, after which PHMG-p was applied to assess its aggravating effect under this pre-stressed condition. This approach reflects real-world situations in which individuals with ocular surface vulnerability may be exposed to environmental toxicants such as PHMG-p. We agree that demonstrating PHMG-p as an independent inducer of DED would require a model where PHMG-p is the sole variable. To address this, we also included PHMG-p single-treatment groups (0.1% and 0.3%) without scopolamine or desiccation stress, as described in Sections 2 (Results) and 4 (Materials and Methods) of the manuscript. These groups were designed to evaluate whether PHMG-p alone can elicit DED-like changes.

Comments 2: In the first result, the author only evaluated the corneal damage of the rats on the 13th day. Please explain the reason for choosing this time point. And only testing the corneal damage for one day is not representative. Periodic testing should be conducted, with daily or at least three times a week testing of corneal damage. The tear fluid accumulation should also be done in the same way.

Response 2: Thanks for comment. In this study, day 13 was selected as the primary endpoint to assess corneal damage and tear parameters because it aligns with the culmination of both the DED induction protocol and PHMG-p exposure period. This time point was chosen based on previous studies demonstrating that DED-related pathological changes—including tear reduction, goblet cell loss, and corneal staining—become stably detectable after 10–14 days of desiccation and scopolamine treatment. While we agree that periodic evaluation could offer a more comprehensive time-course understanding, our primary goal was to assess the cumulative effects of PHMG-p exposure on an already established DED model.

Comments 3: All the charts in the text do not meet the publication requirements. The visualization degree of the horizontal and vertical axes of the column charts is too low.

Response 3: Thanks for comment. The visualization of the horizontal and vertical axes has been adjusted for all charts.

Comments 4: The staining pictures did not add scales, and the background colors were not uniform.

Response 3: Thank you for the comment. We have added scale bars to the staining images. Regarding the background color, we initially observed severe lesions at the 1% PHMG-p concentration, leading us to adjust the highest dose to 0.3% and repeat the experiment. Please note that the variation in background color may be due to slight inconsistencies in staining and imaging timing.

Comments 5: The table arrangement does not conform to the publication requirements. It is recommended to revise it.

Response 5: Thank you for the comment. The table was revised.

Round 2

Reviewer 2 Report

Comments and Suggestions for Authors

Minor revisions:

  1. The group names corresponding to the third and fourth images in Figure 1 A are not standardized and cannot be simply used as 0.1 and 0.3. Please refer to the group names in Figure 3 B for consistency throughout the text.
  2. Figure 2 B, regarding the scale, the size of the first four scales is 100 μm, and the scale of the last two images is 50 μm. Why are they not consistent? Please unify the scale size.
  3. Swap the positions of A and B in Figure 2.
  4. The horizontal line below the “Induced DED” text in Figure 3 B exceeds the length of the image. Please adjust the length of the horizontal line to match the size of the image.
  5. Swap the positions of A and B in Figure 3.
  6. The text in Figures C and D in Figure 5 is not clear. Please upload a clear version.
  7. Figure 6: Unified font for the horizontal and vertical axes of the bar chart. It is recommended to use a unified font for the entire bar chart.
  8. In Figure 9, the figure legend only indicates that * represents p < 0.05, but does not explain the corresponding p values for ** and ***. The caption for the three types of * should be provided in Figure 1 where * first appears.
  9. The text in Figures C and D in Figure 8 is not clear. Please upload a clear version.
  10. There are two formats in the text: p<0.5 and VIP>0.5. It is recommended to unify them and connect them with a space character between the text and<.
  11. There are many details in the images of the article, please carefully correct them to ensure consistency throughout the entire text.
Comments on the Quality of English Language

Language is generally acceptable and expressed relatively clearly, but there may be some issues. Before considering publishing this manuscript, it should be reviewed by a native English speaker or professional editing service.